# Folate–Methionine Cycle Disruptions in ASD Patients and Possible Interventions: A Systematic Review

**DOI:** 10.3390/genes14030709

**Published:** 2023-03-13

**Authors:** Melissa Roufael, Tania Bitar, Yonna Sacre, Christian Andres, Walid Hleihel

**Affiliations:** 1Department of Biology, Faculty of Arts and Sciences, Holy Spirit University of Kaslik, Jounieh P.O. Box 446, Lebanon; 2UMR Inserm 1253 Ibrain, Université de Tours, 37032 Tours, France; 3Department of Nutrition and Food Sciences, Faculty of Arts and Sciences, Holy Spirit University of Kaslik, Jounieh P.O. Box 446, Lebanon

**Keywords:** autism, *MTHFR*, vitamin B12, folate, folate–methionine cycle, homocysteine

## Abstract

Autism Spectrum Disorder (ASD) has become a major public health concern due to its rapidly rising incidence over the past few years. Disturbances in folate or methionine metabolism have been identified in many individuals with ASD, suggesting that the folate–methionine cycle may play an essential role in the pathogenesis of autism. Thus, changes in metabolite concentrations associated with this cycle could be used as potential biomarkers and therapeutic targets for ASD. The aim of this systematic review is to elucidate the perturbations of this cycle and the possible interventions that may be proposed in this context. Several studies have shown that high levels of homocysteine and low levels of vitamins B12 and folate are associated with ASD. These changes in serum metabolites are influenced by poor diet. In fact, children with ASD tend to eat selectively, which could compromise the quality of their diet and result in nutrient deficiencies. Moreover, these disturbances may also be caused by genetic predispositions such as polymorphisms of the *MTHFR* gene. Few studies have demonstrated the beneficial effects of the use of nutritional supplements in treating ASD children. Therefore, larger, well-structured studies are recommended to examine the impact of vitamin B12 and folate supplementation on homocysteine levels.

## 1. Introduction

Autism Spectrum Disorders (ASD) are a group of neurodevelopmental disorders that appear during the early years of development and are characterized by persistent challenges in social communication and interaction, restricted interests, and repetitive behaviors [1]. During the past several years, the number of ASD cases has markedly increased. The prevalence of ASD worldwide is estimated to affect 1 in 100 children, with a male to female ratio of 4:1 [2]. However, the prevalence of ASD in Lebanon is much higher; it is estimated to affect 1 in 66 children in Beirut and Mount Lebanon regions [3].

To date, the exact etiology of ASD remains unclear, but several studies suggest that it is a multifactorial disorder involving a strong genetic component and environmental factors [4]. In fact, it has been shown that chromosomal abnormalities, single nucleotide variants (SNVs), and rare copy number variations (CNVs) are associated with ASD [5].

In a previous study, using high-resolution array-CGH and real-time PCR (qPCR), rare CNVs were uncovered and identified in a group of ASD Lebanese patients. Among the CNVs detected, novel ASD candidate genes *PJA2*, *SYNPO*, *APCS*, and *TAC1* have been identified in this group. In addition, previously described CNVs containing genes such as *SHANK3*, *MBP*, *CHL1*, and others have been identified [6]. Moreover, Soueid et al. assessed the contribution of genetic variation to ASD in the Lebanese population. The study led to the identification of new variants and previously unrecognized susceptibility candidate genes for ASD. In fact, the *RYR2*, *PTDSS1*, and *AREG* genes were uncovered and identified as novel potential autism susceptibility genes [7]. Furthermore, it is proposed that several environmental factors could contribute to the development of this disorder. Among these have been suggested advanced maternal age (35 years), maternal chronic hypertension, pre-eclampsia, gestational hypertension, and overweight before or during pregnancy [8]. A case-control study, conducted between 2015 and 2020 in the Lebanese population, shed light on risk and protective factors associated with ASD. In this study, consanguinity, a familial history of ASD and attention deficit hyperactivity disorder (ADHD), and maternal stress during pregnancy were identified as risk factors associated with ASD [9]. Additionally, perturbations in metabolism may also be present in ASD patients [10]. Several studies suggest that ASD is associated with altered energy metabolism, increased oxidative stress, decreased methylation capacity, and impaired sulfur metabolism [11,12,13]. In fact, disturbances in the folate–methionine cycle have been identified in many individuals with ASD [14,15].

The folate cycle begins with the conversion of dietary folate (vitamin B9) into dihydrofolate (DHF) then tetrahydrofolate (THF) by dihydrofolate reductase (DHFR). Next, THF is converted to 5, 10-methylene-THF by serine hydroxymethyltransferase (SHMT) before being reduced into 5-methyl-THF (5-mTHF) by methylenetetrahydrofolate reductase (MTHFR). As part of the methionine cycle, 5-mTHF donates a methyl group to regenerate methionine (Met) from homocysteine (Hcy), which is catalyzed by methionine synthase (MS) and requires vitamin B12 as a cofactor (Figure 1) [16].

The first step in methionine metabolism is performed by adenosine triphosphate (ATP) and methionine adenosyltransferase (MAT), an enzyme that converts methionine into *S*-adenosyl-methionine (SAM), the body’s main methyl donor. During methylation reactions, SAM donates its methyl group and is converted into *S*-adenosylhomocysteine (SAH). SAH is later hydrolyzed by *S*-adenosyl-L-homocysteine hydrolase to homocysteine and adenosine (Ado), as shown in Figure 1. Homocysteine can also enter the trans-sulfuration pathway, which involves the conversion of homocysteine into cysteine, which is the rate-limiting precursor of glutathione (GSH), the major antioxidant in the body through cystathionine β synthase and vitamin B6 [17]. Methionine synthase, which utilizes vitamin B12 as a cofactor, catalyzes the re-methylation of homocysteine to methionine [18].

This pathway plays a fundamental role in DNA synthesis, DNA methylation, and the regulation of cellular redox homeostasis [19]. Folate plays an important role in pregnancy and fetal development as it is essential for cell division and tissue growth and promotes closure of the neural tube in the uterus. This closure is considered the first step to forming the brain and spinal cord. In the fetus, folate is important for cell proliferation, neural stem cell proliferation and differentiation, reduction of apoptosis, and modification and maintenance of DNA synthesis. Prenatal folate deficiency can lead to altered neurodevelopment, brain health, and neural tube defects in offspring [20]. Therefore, changes in metabolite concentrations associated with this cycle could be used as potential biomarkers and therapeutic targets for ASD.

Herein, we review the changes in metabolite concentrations associated with this cycle, their causes, and the possible interventions that may be proposed in this context.

## 2. Materials and Methods

This systematic review was conducted rigorously following the Preferred Reporting Items for Systematic Reviews and Meta-Analyses (PRISMA) guidelines [21]. We collected relevant data that conformed to the eligibility criteria of our study.

### 2.1. Study Design

According to previously published evidence of the folate–methionine pathway’s involvement in autism, we conducted a systematic review to evaluate the perturbations of the folate–methionine cycle in individuals with ASD, their causes, and the possible interventions.

### 2.2. Eligibility Criteria

#### 2.2.1. Inclusion Criteria

Studies were selected based on the following inclusion criteria: (1) all published randomized controlled trials (RCTs), observational studies (cohort or case-control designs), and reviews dealing with the folate–methionine cycle in ASD; (2) articles published in English, (3) studies that used valid autism diagnostic tools such as The Diagnostic and Statistical Manual of Mental Disorder, 5th edition (DSM-V), (4) studies including data for metabolites, cofactors, or genes of the folate–methionine pathway; (5) interventions using metabolites or cofactors of the folate–methionine pathway.

#### 2.2.2. Exclusion Criteria

The exclusion criteria were as follows: (1) abstracts, conference papers, posters, and in vitro studies, and (2) studies with duplicate or overlapping data. Only data published between 2016 and 2022 were included.

### 2.3. Literature Search and Selection of Articles

We searched in the PubMed, Scopus, and Medline databases using different key words to identify all studies that covered the folate–methionine cycle in autism. The following text words were used in the study: “folate–methionine cycle and autism”, “folate and autism”, “folate deficiency and autism”, “vitamin B12 and autism”, “vitamin B12 deficiency and autism”, “vitamin B6 and autism”, “methionine and autism”, “*MTHFR* polymorphism and autism”, “antibodies FRA and autism”, “folic acid supplementation and autism”, and “vitamin B12 supplementation and autism”.

In the first stage, we reviewed the titles and abstracts based on the inclusion and exclusion criteria. In the second stage, the full text of relevant articles remaining was examined, and irrelevant studies were excluded. To avoid bias, all steps were reviewed by two reviewers independently, and reasons for deleting articles were mentioned. No disagreements were noted between the two authors. Figure 2 reports a schematic diagram of the literature search procedure.

### 2.4. Data Extraction

Seventeen articles met the criteria and were selected for inclusion in our systematic review. Data extracted from each eligible article included the study name; publication year; main outcome; study outcome parameters; sample size; type of study; age of children; and evaluation of folic acid, vitamin B12, and vitamin B6 supplementation used.

### 2.5. Risk of Bias across Studies

Studies examining the levels of homocysteine, folate, and vitamin B12 did not quantify unmeasured factors that might have an impact on the findings, such as the usage of vitamins for more than 3 months previous to the collection of blood samples. Another limitation was the lack of assessment of serum levels of vitamin B12, folate, and their metabolites in the majority of the investigated ASD children and controls. Additionally, the majority of the studies we analyzed used small sample sizes, which raised the possibility of bias. Further studies should be conducted with larger cohorts in various geographic regions and with various ethnic groups. Another problem that must be considered in order to more effectively combine treatment procedures is the absence of trustworthy biomarkers that could accurately predict who might benefit from a given intervention.

## 3. Results

### 3.1. Overall Results

A total of 1461 articles were identified using our search strategy, as shown in Figure 1. We excluded duplicates and other studies that were ineligible for various reasons, such as in vitro studies, reviews, and/or articles that did not support the main discussion. Finally, 17 studies were systematically reviewed.

### 3.2. Summary of the Results Included in This Systematic Review

Studies regarding the deficiency of B vitamins (4 studies), the poor dietary intake of ASD children (3 studies), the genetic predisposition of ASD (7 studies), and the effect of folate and vitamin B12 supplementation in ASD children (3 studies) are summarized in Table 1, Table 2, Table 3 and Table 4.

### 3.3. Perturbations of the Folate–Methionine Cycle in Individuals with ASD

#### 3.3.1. Studies on the Deficiency of B Vitamins in ASD Children

The levels of vitamin B12 and folate were significantly lower in children with ASD than in the control group in most of the studies [40,41]. These deficiencies cause a decrease in homocysteine re-methylation, resulting in an increase in homocysteine levels. A case control study conducted in Saudi Arabia among 70 children with ASD and 85 children without ASD reported a lower intake of vitamin B12 [23]. Similar results were obtained by Esteban-Figuerola et al., indicating lower intake of vitamin B12 among children with ASD [24]. A further study by Kelly Barnhill et al. assessed the dietary intake of 86 children with ASD aged 2–8 years and 57 age-matched peers without ASD. The majority of children with ASD were deficient in a number of B vitamins, including vitamin B6 and folate [25]. Moreover, Julio Plaza-Diaz et al. investigated the nutrient intakes of Spanish preschool children with ASD. They found that the usual individual intakes of vitamins B6, B12, and folate were below the recommended doses in children with ASD when compared with the control children [26].

#### 3.3.2. Causes of These Perturbations

##### Studies on the Poor Dietary Intake of ASD Children

Vitamin B deficiency can be caused by a poor diet. In fact, children with ASD or picky eaters are five times more likely to experience feeding difficulties than children without ASD. They have a tendency to eat selectively, which could compromise the quality of their diet and result in nutrient deficiencies. Food refusal, limited food variety based on type or food texture, increased consumption of particular food items, or rejection of certain foods are all examples of food selectivity [42]. Chistol et al. compared oral sensory processing in children aged 3 to 11 who had ASD (*n* = 53) and those who did not (*n* = 58). They also examined the relationships between atypical oral sensory processing, food selectivity, and fruit/vegetable consumption in children with ASD. The results showed that children with ASD exhibit more atypical sensory processing than children without ASD. They tend to refuse more types of food and consume fewer vegetables than those with typical oral sensory sensitivity [27]. Adams et al. highlighted the food groups that were preferred by children with ASD, demonstrating a preference for starches and snack items compared to fruits and vegetables [28]. Moreover, a cross-sectional case-control study was carried out in 144 children (*N* = 55 with ASD; *N* = 91 with neurotypical children) between 6 and 18 years of age. Body composition, nutritional intake, food consumption frequency (FFQ), and mealtime behavior were evaluated. Children with ASD showed an unbalanced body composition toward both underweight and obesity, a greater degree of inadequate intake, high food selectivity as shown by their consumption frequency, and more disrupted eating behavior [29].

##### Studies on the Genetic Predisposition of ASD

The etiological factors leading to vitamin B deficiency and elevated homocysteine levels in the body may not only be due to poor diet but may also be caused by genetic predispositions. For instance, mutations and polymorphisms in metabolic enzymes such as methylenetetrahydrofolate reductase MTHFR have been linked to vitamin B12 deficiency [43]. This enzyme is responsible for maintaining methionine and homocysteine balance to prevent cellular dysfunction. It is involved in converting 5,10-methylenetetrahydrofolate to 5-methyltetrahydrofolate, which is required for the conversion of homocysteine to methionine by methionine synthase, a B12 dependent enzyme [44]. Decreased activity of the MTHFR enzyme may alter DNA synthesis and methylation and increase plasma homocysteine levels. High homocysteine levels and increased oxidative stress are commonly associated with the pathophysiology of many neuropsychiatric disorders, including autism spectrum disorders [45]. Many polymorphisms of the *MTHFR* gene have been discovered. Among them, C677T and A1298C are two common polymorphisms that have been established to reduce enzyme activity. The *MTHFR* C677T mutant is due to a cytosine change in thymine at position 677 of Exon 4, where alanine is replaced by valine, resulting in a thermolabile enzyme with reduced enzyme activity. For instance, the homozygous C677T (TT) condition is associated with increased Hcy and lower folate levels. Additionally, the *MTHFR* A1298C mutant is due to an adenine change in a cytosine at position 1298 of exon 7. This change leads to the replacement of glutamic acid by an alanine and results in producing a more detrimental reduced activity in the homozygous 1298CC condition [46,47].

Several studies have mainly focused on the influence of the *MTHFR* C677T and A1298C on autism susceptibility, but the findings are still inconclusive. Chen-Xi Li et al. assessed the association between *MTHFR* C677T polymorphism and the risk of autism in the Chinese Han population. The results suggested that *MTHFR* C677T was strongly associated with the increased risk of autism in China, especially in the Northern Han subgroup [30]. A case control study, conducted among an Egyptian sample of patients with autism, was aimed at the identification of possibly existing C677T and A1298C polymorphisms within the *MTHFR* gene. The findings proposed a significant association between severity and occurrence of autism with *MTHFR* gene polymorphisms C677T and A1298C. In fact, heterozygosity for A1298C polymorphism was highest among patients (41.9%) followed by 35.5% mutant genotype CC and 22.6% normal AA (wild) type, and Allele C was detected in patients more than in the control (56.45% vs. 11.54%) (*p* < 0.001). For C667T polymorphism, patients had the highest heterozygosity (48.4%) compared to wild type genotypes CC (38.7%) and mutant genotypes TT (12.9%), and Allele T was detected more in patients than in controls (31.10% vs. 5.13%) (*p* < 0.00) [31]. Furthermore, heterozygosity for CT and AC genotypes were detected equally among patients with severe autism. In addition, Al-Omari et al. tested the serum levels of oxidant/antioxidant status along with analysis of *MTHFR* C677T polymorphism in Jordanian autistic children. The results showed that the serum levels of cysteine, folate, and vitamin B12 in autistic patients were significantly lower than in the control group. In addition, high levels of serum homocysteine were found in children with autism compared to the control group. The frequency of *MTHFR* C677T was significantly higher in children with autism than in the control group. The homozygous genotype CC of the *MTHFR* C677T was lower in patients with autism than in the control group (28% vs. 52%), whereas the heterozygous CT and homozygous TT genotypes were higher in patients with autism compared to the control group (52%, 44%) and (20%, 4%), respectively [32].

In contrast, a meta-analysis showed that *MTHFR* C677T polymorphism is remarkably associated with ASD in the five genetic models (allelic, dominant, recessive, heterozygote, and homozygote). However, *MTHFR* A1298C polymorphism was not found to be significantly associated with ASD in the five genetic models [33,34]. Concordant observations were made in addition by Bahman Razi et al. The authors suggested that there is a strong association between *MTHFR* C677T gene polymorphism and autism spectrum disorders risk in Caucasians, but there is no significant association between *MTHFR* A1298C gene polymorphism and ASD risk [35]. Moreover, Muftin et al. examined the correlation between common polymorphism (A1298C) and the risk of autism on Iraqi autistic children. The results showed that all three alleles (AA, AC, CC) were non-significantly correlated with the risk of autism (O.R = 1.23, *p* = 0.75) and that the A1298C polymorphism is not associated with autism in the Iraqi population [36].

The association between *MTHFR* polymorphisms and ASD is still inconclusive; therefore, further larger and well-structured studies are recommended.

#### 3.3.3. Possible Interventions

##### Studies on the Effect of Folate and Vitamin B12 Supplementation in ASD Children

Folic Acid Supplementation

Folic acid is the acid form of folate, and its intake may decrease the levels of homocysteine and regulate the perturbations of the folate cycle. There are few published articles that have evaluated the effects of folic acid intervention on the treatment of autistic children. However, an open-label trial tested the dose of folic acid for a period of three months and found significant improvements. In that study, 44 autistic children received 400 μg of folic acid twice daily in an open-label experiment, while 22 control children received no supplementation. Additionally, blood was taken from 29 typically developing children and 29 autistic children, who were randomly chosen from the intervention group. The study showed that taking supplements of folic acid reduced the symptoms of autism in terms of sociability, cognitive verbal/preverbal, receptive language, and affective expression and communication. Moreover, this treatment also increased the concentrations of folic acid, homocysteine, and normalized glutathione redox metabolism [37].

Vitamin B12 Supplementation

Several studies have demonstrated the beneficial effects of the use of nutritional supplements in treating ASD children. Because vitamin B12 is a vital cofactor for regenerating methionine from homocysteine, it has been used for enhancing methylation capacity and improving the redox status of children with ASD [48]. Adela Corejová et al. examined the efficiency of a syrup form of methylcobalamin in treating autism; 500 µg of methylcobalamin was administered daily to autistic children and young adults during a 200-day period. The results suggested that methylcobalamin treatment increased the levels of vitamin B12 in ASD children, but no significant changes were observed in homocysteine levels. Moreover, methylcobalamin treatment had an important impact on the overall oxidative status, expressed as GSH/GSSG [38]. In another study, a total of 57 children with ASD were randomly assigned to 8 weeks of treatment with methyl B12 (75 μg/kg) or saline placebo every 3 days in a subcutaneous injection. The study showed an increase in plasma methionine (*p* = 0.05), a decrease in *S*-adenosyl-L-homocysteine (SAH) (*p* = 0.007), and an improvement in the ratio of *S*-adenosylmethionine (SAM) to SAH (*p* = 0.007), thus leading to the improvement of cellular methylation capacity [39].

## 4. Discussion

Vitamin B12 and folate, the two main key regulators of the folate–methionine cycle, are required for homocysteine metabolism. Hence, any defect in them leads to homocysteine accumulation in the body, thus leading to neurotoxicity, oxidative stress, and mitochondrial dysfunction [49,50]. In fact, several studies reported that children with ASD had lower levels of vitamin B12 and folate and higher levels of homocysteine compared with healthy controls.

Vitamin B12, a water-soluble vitamin, plays an important role in normal brain functioning. It is required for the development and initial myelination of the central nervous system, and its deficiency is associated with developmental delays, irritability, weakness, and failure to thrive [51]. Vitamin B12 plays a vital role in DNA synthesis and cellular energy production [52]. A number of symptoms specific to gastrointestinal, hematological, neurological, and psychiatric disorders can be used to diagnose its insufficiency. Some of the well-known symptoms of vitamin B12 deficiency include neurological impairments such as motor disturbances, abnormal balance and reflexes, sensory and memory loss, cognitive impairment, irritability, and cerebral atrophy [53]. Moreover, vitamin B12 is a vital cofactor for the remethylation of homocysteine to methionine by providing methyl groups for the transmethylation and transsulfuration metabolic pathways. Lack of vitamin B12 may lead to increased concentrations of homocysteine and decreased concentrations of methionine and *S*-adenosylmethionine (SAM). Decreased concentrations of SAM might moreover influence the transsulfuration pathway, leading to reduced levels of cysteine and GSH, and thus reduced antioxidant capacity [54]. Inconsistencies were found between studies in plasma concentrations of metabolites and/or cofactors of the folate–methionine pathway. These changes in metabolite concentrations may be caused by abnormalities in folate transport and/or metabolism [17]

Folate is a naturally occurring form of vitamin B9, which people aquire from their diet. It can be obtained from dietary sources such as beans, nuts, meat, leafy vegetables, or by supplementing it in the form of folic acid [55]. Folate is considered to be an important key factor during neurodevelopment. It is required to carry one-carbon groups for methylation reactions, among them nucleotide base synthesis. It is essential for DNA replication and repair, as well as for RNA synthesis. It is also involved in homocysteine remethylation reactions. Therefore, folate is vital during early embryogenesis, which is characterized by rapid cell division. Moreover, it is required for amino acid metabolism, neurotransmitters, and phospholipid biosynthesis [56]. As a result, folate plays an important role in the development and function of the central nervous system. Moreover, folic acid has been found to be essential for rapid tissue growth and cell division during fetal development. In fact, several studies have focused on the importance of folic acid in improving behavioral outcomes in children and its role as a modifiable risk factor for ASD [22].

Inadequate dietary intake and genetic polymorphisms of the folate–methionine cycle may cause vitamin B deficiency. In fact, several studies reported that children with ASD exhibit selective feeding five times more frequently than typically developing children [57]. They have been found to have certain feeding difficulties, mainly food selectivity, food refusals, and disruptive eating behaviors [58]. Consequently, these common feeding problems result in nutritional deficiencies and malnutrition [59]. Moreover, perturbations in the folate–methionine cycle may not only be due to poor diet, but may also be caused by genetic predispositions. Methylenetetrahydrofolate reductase (MTHFR) is one of the most important enzymes in the folate pathway. Single nucleotide polymorphisms (C677T and A1298C) were reported to be associated with the decline in *MTHFR* activity, thus leading to an accumulation of homocysteine. However, the association between *MTHFR* C677T/A1298C and susceptibility to autism spectrum disorders is still debatable. Several studies reported a strong association between *MTHFR* C677T gene polymorphism and autism spectrum disorders risk, while polymorphism was not associated with autism risk overall [35,36]. Therefore, further larger and well-structured studies are recommended.

Few studies evaluated the effect of vitamin B12 and folate supplementation in ASD-diagnosed children. Sun et al. reported that taking folic acid supplements reduced the symptoms of ASD, elevated levels of homocysteine and folic acid, and normalized glutathione redox metabolism [37]. Regarding the studies that have investigated the vitamin B12 efficacy in ASDs children, Corejová et al. reported that methylcobalamin treatment increased the levels of vitamin B12, and Hendren et al. reported that Methyl B12 treatment lead to an improvement of cellular methylation capacity [39]. A case report study conducted by Fadila et al. showed the effect of a three-month folic acid supplementation in a two-year-old child with a heterozygous *MTHFR* genetic polymorphism. The results revealed an enhancement in speech and general behavior [60]. Moreover, Gowda et al. evaluated the impact of betaine, pyridoxine, folic acid, and vitamin B12 supplementation on the lowering of total homocysteine concentrations. A five-year-old child with a heterozygous *MTHFR* genetic polymorphism was used in the study. Serum homocysteine levels were high at 160.30 (5–15 mmol/L) prior to therapy. Significant improvement was seen following treatment, with serum homocysteine levels falling to 78 mmol/L [61]. In addition, it has been shown that supplementation of folic acid and betaine improved plasma concentrations of metabolites in the methionine pathway, and vitamin B12 supplementation further enhanced these concentrations. On the other hand, supplementation with folinic acid given to children with low-functioning autism and at least one symptom of CFD led to an improved cerebral folate status and remarkable cognitive, motor, and neurologic changes [17]. Moreover, supplementation with folic acid and/or folinic acid in autistic children has improved several neurological and behavioral symptoms as well as the concentration of one-carbon metabolites [22]. Further research is also needed to determine the long-term effect of folate and vitamin B12 supplementation on the symptoms of ASD, as well as their impact on the levels of homocysteine, vitamin B12, and folate, taking into account a larger sample.

## 5. Conclusions

In summary, this review aimed to evaluate the perturbation of the folate–methionine cycle in ASD and the possible interventions. Several studies have shown that high levels of homocysteine and low levels of vitamin B12 and folate are associated with ASD. These changes in serum metabolites are influenced by poor diet or genetic polymorphism in the *MTHFR* gene. Although many dietary interventions have been studied, a decisive recommendation for a specific nutritional therapy as a standard treatment for ASD cannot be made due to the lack of conclusive scientific data regarding the effect of therapeutic diets on ASD. Therefore, larger and well-structured studies are recommended to evaluate the association between *MTHFR* polymorphism and ASD. Additionally, there is a need to examine the effects of vitamin B12 and folate supplementation on homocysteine levels.

## Figures and Tables

**Figure 1 genes-14-00709-f001:**
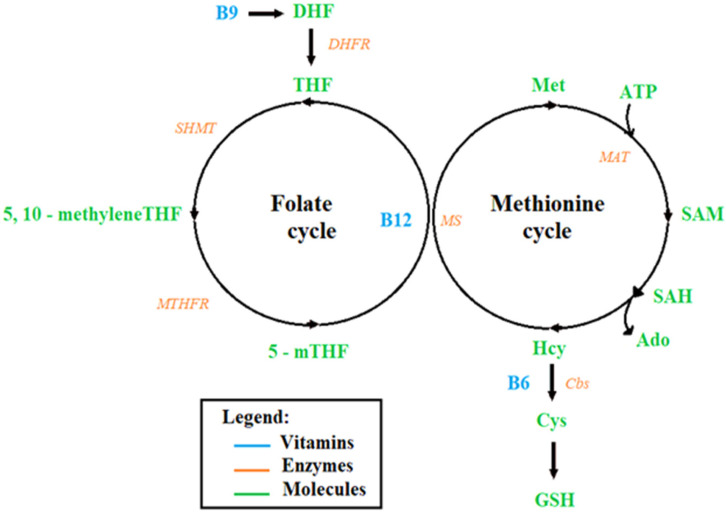
Folate and methionine cycle [16]. Ado adenosine, ATP adeno-sine triphosphate, vitamin B6, vitamin B9, vitamin B12, Cbs cystathionine β synthase, DHF dihydrofolate, DHFR dihydrofolate reductase, Cys cysteine, GSH glutathione, Hcy homocysteine, MAT methionine adenosyltransferase, Met methionine, 5–methylTHF, 5, 10–methyleneTHF, MTHFR methylenetetrahydrofolate reductase, MS methionine synthase, SAM *S*-adenosyl-methionine, SAH *S*-adenosylhomocysteine, SHMT hydroxymethyltransferase, THF tetrahydrofolate.

**Figure 2 genes-14-00709-f002:**
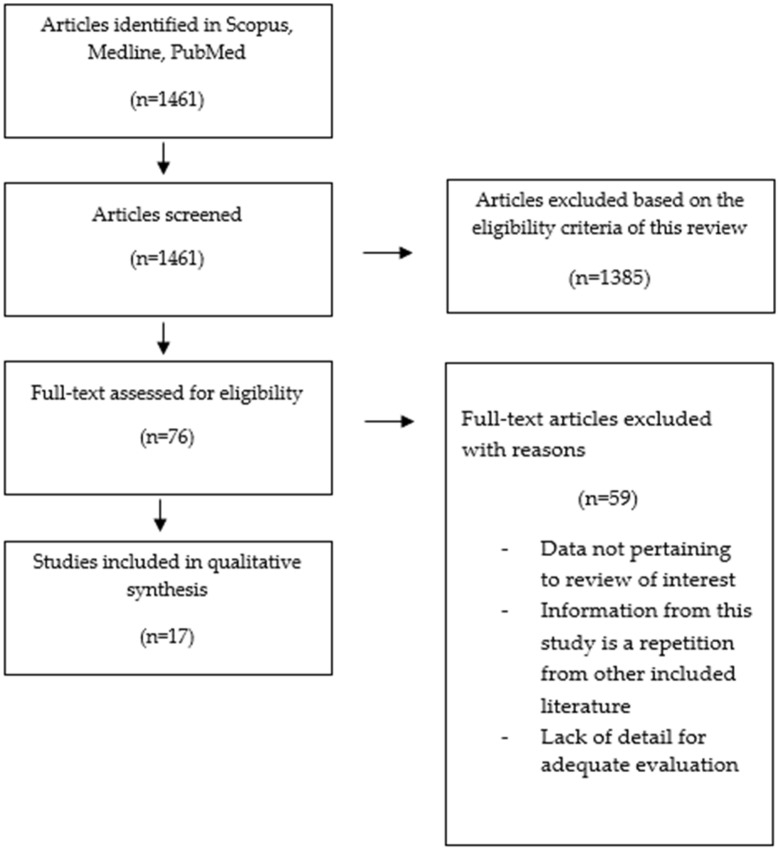
PRISMA flow diagram: schematic diagram of literature search and selection for articles included in this systematic review [22].

**Table 1 genes-14-00709-t001:** Studies on the deficiency of B vitamins in ASD children.

No	Study	Type of Study	Sample Size	Age	Aim of the Study	Study Outcomes Parameters	Outcomes
1	S. Alahmari et al. (2022) [23]	Case control study	155 children (70 children with ASD and 85 children without ASD)	7–12 years old	Compare the nutritional status of Saudi children with autism spectrum disorders (ASD) to Saudi children without ASD	Dietary consumption using a 3-day food record (daily average of energy, intake of folic acid, vitamin B12, calcium, iron, magnesium, sodium, selenium vitamin B6, vitamin C)	The daily average of energy from carbohydrates was significantly higher among children with ASD than among children without ASD. A lower intake of folic acid, vitamin B12, calcium, iron, magnesium, sodium, and selenium was detected among children with ASD, while vitamin B6, vitamin C, and potassium were significantly higher among children with ASD compared to children without ASD [23].
2	Esteban-Figuerola et al. (2020) [24]	Meta-analysis	-	-	Determine the overall differences in nutritional intake and food consumption between children with autism spectrum disorder and control (typical development) children	Food record (calcium, vitamin B12, vitamin D, vitamin E, calcium, selenium, phosphorus, thiamine, riboflavin	Children with ASD consume less protein, vitamin D, vitamin B12, omega-3, calcium, phosphorus, selenium, thiamine and riboflavin than controls [24]
3	K Barnhill et al. (2018) [25]	Case control study	86 children with ASD aged 57 age-matched peers without ASD.	2–8 years old	Compare dietary intake in 2-to-8-year-old US children with and without an ASD diagnosis	Food record (energy, protein, carbohydrate, vitamin B6, vitamin B, vitamin A, vitamin D, folic acid)	Children with ASD consumed lower levels of protein and calcium and were deficient in a number of B vitamins, including B1, B2, B3, B6, and folate, compared to similarly-aged children without ASD [25]
4	J Plaza-Diaz et al. (2021) [26]	Case control study	54 children with ASD and 57 typically developing children	2–6 years old	Determine the feeding behavior, dietary patterns, and macro-and micronutrient intakes in a sample of Spanish preschool children with ASD compared to typically developing control children of the same age	Food frequency questionnaire and 24 h dietary registrations	High energy and fat intakes and a low intake of vegetables and fruits. The usual individual intakes of vitamins B6, B12, and folate were below the recommended doses in children with ASD when compared with the control children [26]

**Table 2 genes-14-00709-t002:** Studies on the poor dietary intake of ASD children.

No	Study	Type of Study	Sample Size	Age	Aim of the Study	Study Outcomes Parameters	Outcomes
1	Chistol et al. (2018) [27]	Cross-sectional study	53 children with ASD and 58 children without ASD	3–11 years old	Compare oral sensory processing function between children with and without ASD, examine the relationship between atypical oral sensory processing and food selectivity in children with ASD, and examine the relationship between atypical oral sensory processing and fruit and vegetable consumption	Food frequency questionnaire and 3 days food record	Children with ASD exhibit more atypical sensory processing than children without ASD. They tend to refuse more types of food and consume fewer vegetables than those with typical oral sensory sensitivity [27]
2	Adams et al. (2022) [28]	Cross-sectional quantitative research design	40 caregivers of ASD children	3–10 years old	Determine the types of feeding difficulties prevalent in children with ASD, the food items that children in South Africa prefer, and the relationship between age and ASD severity on food preferences	Behavioral Pediatric Feeding Assessment Scale and food preference questionnaire	Common feeding difficulties in children with ASD. A preference for starches and snack items compared with fruits and vegetables. Significant correlations between ASD severity and ASD age on food preferences [28]
3	J Molina-López et al. (2021) [29]	Cross-sectional case-control study	144 children (*N* = 55 with ASD; *N* = 91 with neurotypical children)	6–18 years old	Evaluate body composition, nutritional status through food selectivity and degree of inadequate intake, and mealtime behavior in children with autism spectrum disorder (ASD) compared to neurotypical children.	Body composition, nutritional intake, food consumption frequency (FFQ), and mealtime behavior	A greater presence of children with a low weight and obesity in the ASD group. The presence of obesity in ASD children compared to the comparison group was even higher when considering the fat component. ASD children had greater intake inadequacy, high food selectivity by FFQ, and more eating problems (food rejection, limited variety, disruptive behavior), compared to neurotypical children [29]

**Table 3 genes-14-00709-t003:** Studies on the genetic predisposition of ASD.

No	Study	Type of Study	Sample Size	Age	Aim of the Study	Study Outcomes Parameters	Outcomes
1	Chen-Xi Li et al. (2021) [30]	Meta-analysis	-	-	Evaluate the association of *MTHFR* C677T polymorphism with autism susceptibility among a Chinese Han population	Genetic analyses: *MTHFR* C677T as predictors of autism risk.	*MTHFR* C677T was strongly associated with the increased risk of autism in China, especially in Northern Han subgroup [30]
2	F. El-baz, et al. (2017) [31]	Case control study	31 ASD children and 39 TD children	1.5 to 18 years	Identify C677T and A1298C polymorphic genotypes of *MTHFR* gene among Egyptian ASD children	Identification of C677T and 1298AC polymorphic genotypes of *MTHFR* gene	There is a significant association between severity and occurrence of autism with *MTHFR* gene polymorphisms [31]
3	L. M. Al-Omari et al. (2020) [32]	Case control study	25 ASD children and 25 TD children	7 to 18 years	Investigate *MTHFR* C677T polymorphism (rs1801133) in autistic children	Frequency of genotype *MTHFR* C677T in children	*MTHFR* C677T frequency was significantly higher in autistic as compared to non-autistic children. The homozygous genotype CC of the *MTHFR* C677T was lower in patients with autism than in the control group (28% vs. 52%), while heterozygous CT genotype of the *MTHFR* C677T and the homozygous TT genotype were higher in patients with autism compared to control group (52%, 44%) and (20%, 4%), respectively [32]
4	Y. Li et al. (2020) [33]	Meta-analysis		-	Authenticate correlations between *MTHFR* polymorphism (C677T/A1298C) and susceptibility to ASD	Genetic analyses: *MTHFR* C677T, *MTHFR* A1298C as predictors of autism risk.	*MTHFR* C677T polymorphism is a susceptibility factor for ASD, and *MTHFR*. A1298C polymorphism is not associated with ASD susceptibility [33]
5	T. Sadeghiyeh et al. (2019) [34]	Systematic and meta-analysis	-	-	Evaluate association of *MTHFR* 677C > T and 1298A > C polymorphisms with risk of autism	Genetic analyses: *MTHFR* C677T, *MTHFR* A1298C as predictors of autism risk.	*MTHFR* 677C > T polymorphism was significantly associated with an increased risk of autism in overall population and by ethnicity, while *MTHFR* 1298A > C polymorphism was not associated with autism risk overall [34]
6	B. Razi et al. (2020) [35]	Systematic review and meta-analysis	-	-	Assess candidate genes associated with ASD risk	Genetic analyses: *MTHFR* C677T, *MTHFR* A1298C as predictors of autism risk.	A strong association between *MTHFR* C677T gene polymorphism and autism spectrum disorders risk in Caucasians. There is no significant association between *MTHFR* A1298C gene polymorphism and ASD risk [35]
7	N. Q. Muftin et al. (2020) [36]	Review	-	-	Examined the correlation of the common polymorphism (A1298C) and risk of autism on Iraqi autistic children	The correlation of the common polymorphism (A1298C) and risk of autism	All three alleles (AA, AC, CC) were non-significantly correlated with the risk of autism (O.R = 1.23, *p* = 0.75) and the A1298C polymorphism is not associated with autism in Iraqi population [36]

**Table 4 genes-14-00709-t004:** Studies on the effect of folate and vitamin B12 supplementation in ASD children.

No	Study	Type of Study	Sample Size	Age	Supplement Used	Period of Intake	Study Outcomes Parameters	Outcomes
1	C. Sun, et al. (2016) [37]	Open-Label Trial	66 ASD children and 22 TD children	4.5 ± 1.1 years old	Folic acid tablets (400 µg) ren	Twice daily/3 months	There is a significant association between severity and occurrence of autism with *MTHFR* gene polymorphisms C677T and A1298C. Further studies are needed on a larger scale to explore other gene polymorphisms that may be associated with autism to correlate the genetic basis of autism. There is a significant association between the severity and occurrence of autism with *MTHFR* gene polymorphisms. C677T and A1298C. Further studies are needed on a larger scale to explore other gene polymorphisms that may be associated with autism to correlate the genetic basis of autism. Plasma levels of FA homocysteine glutathione metabolism before and after treatment. Improvement of autism symptoms	Taking supplements of folic acid reduced the symptoms of autism in terms of sociability, cognitive verbal/preverbal, receptive language, and affective expression and communication. It increased the concentrations of folic acid, homocysteine, and normalized glutathione redox metabolism [37]
2	A. Čorejová et al. (2022) [38]	Case control study	25 ASD children	4–20 years old	Methylcobalamin syrup (500 µg dose)	Daily/200 days	Levels of vitamin B12 and homocysteine after supplementation. Impact on the overall oxidative status	Methylcobalamin treatment increased the levels of vitamin B12 in ASD children, but no significant changes were observed in homocysteine levels. Moreover, methylcobalamin treatment had an important impact on the overall oxidative status, expressed as GSH/GSSG. Methylcobalamin treatment increased the levels of vitamin B12 in ASD children, but no significant changes were observed in homocysteine levels. Methylcobalamin treatment had an important impact on the overall oxidative status, expressed as GSH/GSSG [38]
3	R. L. Hendren et al. (2016) [39]	Randomized, Placebo-Controlled Trial	57 ASD children	3–7 years	Methyl B12 (75 μg/kg) or saline placebo every 3 days in a subcutaneous injection	8 weeks of treatment	Plasma Plasma levels of methionine, *S*-adenosyl-L-homocysteine, and the ratio of *S*-adenosylmethionine (SAM) to SAH	An increase in plasma methionine (*p* = 0.05), a decrease in *S*-adenosyl-L-homocysteine (SAH) (*p* = 0.007), and an improvement in the ratio of *S*-adenosylmethionine (SAM) to SAH (*p* = 0.007), thus leading to the improvement of cellular methylation capacity [39]

## Data Availability

Not applicable.

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
