# Peer review of "Folate–Methionine Cycle Disruptions in ASD Patients and Possible Interventions: A Systematic Review"

_genes, 2023, doi:10.3390/genes14030709_

Round 1

Reviewer 1 Report

I thank the opportunity to evaluate this nice review manuscript, which will certainly contribute to the current knowledge and practice in the care of patients with ASD. Some points should be evaluated by the authors: 

1. I suggest authors to present all genes cited in the manuscript in italics, as this is a formal recommendation from several societies. 

2. Other suggestion is to use along the text only folate, because it makes quite confusing for the reader why in some cases vitamin B9 is mentioned and why in other paragraphs folate. My suggestion according to the manuscript title is to use only folate along the text. 

3. In Figure 1, my suggestion is to introduce the abbreviations in alphabetic order to make it easier for the reader to localize the meaning of the several abbreviations used in the picture. 

4. Due to the growing use of genomic screening during neonatal and infancy periods, do the authors consider that children with the genetic finding of polymorphisms previously associated with higher risk for ASD should be actively and closely evaluated by the neuropediatrician and possibly treated (with the purpose of preventing the development of severe compromise)?

Author Response

We appreciate the time and effort that the reviewer has dedicated to providing his valuable feedback on our manuscript. We are grateful to the reviewer for his insightful comments on our submitted paper. 

We have been able to incorporate changes to reflect the suggestions provided by the reviewer. 

Reviewer 2 Report

This work is a systematic review on Folate-Methionine cycle disruptions in ASD patients and possible interventions. The review was conducted using the Preferred Reporting Items for Systematic Reviews and Meta-Analyses (PRISMA) guidelines. 

Overall, the paper is very well written and the methodology, results, and conclusions are all clearly stated. 

Regarding the systematic review, the paper yields a clear, complete, and accurate account of the motivation for the review, what the protocol was, and what were the conclusions. 

The only aspect where I feel that the paper could be improved is on a better contextualization of the results, compared with previous work on this topic. In particular regarding other systematic reviews, such as Main 2010 [ref 17] or the work of Hoxha 2021 [ref 22]. 

Both these works are referenced in the text, however, not as much as they should, given that these constitute two previous applications of this approach to the same area, and I feel that this is not sufficiently discussed.

Author Response

(The authors gave the same response as above.)
